# Can Biomass Mastication Assist the Downstreaming of Polyhydroxyalkanoates Produced from Mixed Microbial Cultures?

**DOI:** 10.3390/molecules28020767

**Published:** 2023-01-12

**Authors:** Hiléia K.S. Souza, Mariana Matos, Maria A.M. Reis, José A. Covas, Loïc Hilliou

**Affiliations:** 1Institute for Polymers and Composites, Campus de Azurém, University of Minho, 4800-058 Guimarães, Portugal; 2Associate Laboratory i4HB–Institute for Health and Bioeconomy, NOVA School of Science and Technology, NOVA University Lisbon, 2829-516 Caparica, Portugal; 3UCIBIO—REQUIMTE, Applied Molecular Biosciences Unit, Department of Chemistry, NOVA School of Science and Technology, NOVA University Lisbon, 2829-516 Caparica, Portugal

**Keywords:** polyhydroxyalkanoates, extraction, digestion, bioplastic, mechanical pretreatment, rheology

## Abstract

Polyhydroxyalkanoates (PHAs) are natural polyesters which biodegrade in soils and oceans but have more than double the cost of comparable oil-based polymers. PHA downstreaming from its biomass represents 50% of its overall cost. Here, in an attempt to assist downstreaming, mastication of wet biomasses is tested as a new mechanical continuous biomass pretreatment with potential for industrial upscaling. Downstreaming conditions where both product recovery and purity are low due to the large amount of treated wet biomass (50% water) were targeted with the following process: extraction of 20 g in 100 mL solvent at 30 °C for 2 h, followed by 4.8 h digestion of 20 g in 0.3 M NaOH. Under the studied conditions, NaOH digestion was more effective than solvent extraction in recovering larger PHA amounts, but with less purity. A nearly 50% loss of PHA was seen during digestion after mastication. PHAs downstreamed by digestion with large amounts of impurities started to degrade at lower temperatures, but their melt elasticity was thermally stable at 170 °C. As such, these materials are attractive as fully PHA-compatible processing aids, reinforcing fillers or viscosity modifiers. On the other hand, wet biomass mastication before solvent extraction improves PHA purity and thermal stability as well as the melt rheology, which recovers the viscoelasticity measured with a PHA extracted from a dried biomass.

## 1. Introduction

Polyhydroxyalkanoates (PHAs) are a family of natural polymers which are biosynthesized as energy and carbon storage granules with size of 0.2 to 0.5 micron [1] in the cells of many naturally occurring microorganisms. In contrast to other natural and abundant polymers, such as cellulose or starch, PHA are thermoplastic polyesters just like poly(lactic acids) (PLAs). In contrast to PLA, which requires industrial composting for biodegradation as defined by established standards [2], PHA is a more attractive biosourced polyester, as it naturally biodegrades in land, fields or the ocean [3]. However, by 2021, PHA production is a tenth of PLA production, and it remains a very marginal player in applications such as plastic packaging [4]. Plastic packaging is indeed a relevant application example for biosourced and biodegradable polymers: in the OECD countries, 31% of the plastics consumption is for packaging, which generates 40% of the total plastic waste [5], whereas packaging represents 50% of the biodegradable plastics market [4].

PHAs exhibit properties that hamper their acceptance by the plastic industry. First, PHAs are rather brittle, as they break at strains as low as 1 to 20% depending on their chemical structures [6,7], which prevents them from being used in many flexible packaging applications. Second, industrial polymer processing technologies require specific melt properties, such as viscosity and melt strength, in order to yield products with sufficient performance [8]. PHA melt viscosity is low, which suits injection molding of, for example, rigid packages such as jars, but its melt strength is not large enough to withstand more rheologically demanding processes such as extrusion film blowing [9]. More importantly, the PHA market price is two to five times the price of polyethylene [10,11], which makes nearly 50% of plastic packaging waste [5].

The recently established pilot scale production of PHA based on mixed microbial cultures (MMC) fed with carbon-rich industrial wastes (see [12] and references therein) gives good prospects for production cost reductions when compared to pure cultures grown on specific clean substrates. However, the extraction of MMC PHA from the fermented biomass takes a big share of the overall production cost. For instance, a technoeconomic analysis of the downstreaming of MMC polyhydroxybutyrate (PHB) produced from wastewater calculated that the extraction step can be two to four times more expensive than the fermentation step, contributing to a total production cost ranging from 1.5 to 2 €/kg PHB [13]. Therefore, the downstreaming of MMC PHA has received growing interest, as illustrated by the number of reviews published in the literature over the last couple of years (see e.g., [14,15,16,17]).

Among the various MMC PHA downstreaming processes that have been investigated, two are highlighted in Table 1: solvent extraction and NaOH digestion of the biomass. In solvent extraction, chemicals are used to weaken the cells’ membrane and then solvate the PHA granules. The obtained PHA-rich liquid phase is then separated from the solid cell residues, and PHA is eventually recovered either by solvent evaporation or precipitation of the PHA solution into an antisolvent. Extraction with CHCl_3_ has become a benchmark for evaluating new PHA downstreaming processes in laboratories because it guarantees the recovery of nearly pure MMC PHA (from 82.5 to 100% purity; see Table 1) with large molecular mass (no PHA degradation) from nearly all kinds of microbial strains [18]. However, because concentrated PHA solutions in CHCl_3_ show large viscosities [19], a small amount of biomass must be treated in large volumes of solvents for achieving large recovery yields, as shown in Table 1. The water content in fresh biomasses also significantly lowers the solvent quality of PHA, which may explain a biopolyester recovery yield as low as 18% reported for wet biomasses containing 25 wt% PHA and treated with CH_2_Cl_2_ [20]. Indeed, very few studies targeted the enhancement of CHCl_3_ extraction of MMC PHA using, for example, mechanical pretreatment of biomasses [21,22]. This is perhaps because extraction with chlorinated solvents is usually tagged as not suitable for industrial PHA production [14,15,16,17,18]. NaOH digestion of *Ralstonia eutropha* biomasses with rich PHA content has long been recognized as a PHA downstreaming route with industrial relevance [23]. However, this approach has been poorly tested for the recovery of MMC PHA, as seen in Table 1. A possible explanation for this is that cell breakage or hydrolysis in MMC is thought to be more difficult to achieve than in pure cultures [11,20] where cells can be selected for their fragility and high PHA content. Nevertheless, one may build on previous work performed on pure cultures treated with NaOH before continuous mechanical disruption [24] in an attempt to increase both the PHA purity and the recovery yield reported so far for MMC biomasses containing lower amounts of PHA [25,26,27].

Here, the mastication of fresh MMC before biomass digestion by NaOH was tested. This continuous and upscalable process has never been used for the mechanical disruption of wet PHA biomasses before PHA recovery [14,15,16,17,18]. Furthermore, a recent evaluation of the environmental performances of different PHA downstreaming processes showed that mechanical disruption and the use of NaOH are more promising [36]. The aim of this study was to explore mastication as a new route to enhance the fragility of the cells, thereby facilitating the digestion of the non-PHA material by NaOH. To do this, a biomass containing 50% (weight basis) of MMC PHA produced from fruit waste at pilot scale was selected. The masticating of biomasses with different water contents was first tested for evaluating the feasibility of a continuous mastication process. Then, wet biomasses were masticated with NaOH before digestion. Next, the downstreaming performance of this new approach was compared with the performance of a NaOH digestion of the dried biomass. Additional extractions with CHCl_3_ were performed for two purposes: (1) to understand the impact of using wet biomass, mastication or both on downstreaming performance and (2) to benchmark the MMC PHA’s properties. Polymer characterization was focused on the thermal and melt properties in view to assess the melt processability of the produced biopolyesters. Results from this explorative study suggest that, although mastication does not assist the digestion of the specific biomass tested here, it permits the solvent extraction of PHAs with improved thermal and rheological properties.

## 2. Results and Discussion

### 2.1. Rheological Assessment of Biomass Mastication Ability

The purpose of the masticating unit used in the present study was to process solid-like materials. The screw and squeezing chamber dimensions, as well as the screw profile of the masticating unit, were thus designed to process solids. Therefore, the PHA enriched biomass should exhibit specific rheological properties to be efficiently masticated. Preliminary trials were performed with dried biomass (overnight at 60 °C in air ventilated oven) mixed with different amounts of distilled water in order to screen for the processability of the feeding material. Indeed, dried biomass cannot be processed in the masticating unit, as the powder could not be conveyed by the screw in the feeding zone. The best results were achieved with a feed consisting of 50 g dried biomass mixed with 50 mL water (50 wt% feed). The proper transport of material by the screw was limited either by larger biomass content because the material would stick to the barrel wall, or by smaller solid content, as the liquid would not flow through the squeezing zone at the tip of the screw. Figure 1 displays the rheological properties of the feed that showed the best processability, together with a feed formulated with 30 wt% dried biomass that could not be processed.

Both feeds displayed a viscoelastic and predominantly solid character. Both mechanical spectra showed a storage modulus G’ larger than the loss modulus G” over the whole range of tested frequencies and with a pronounced frequency dependence for G’. The feed with 50 g/50 mL biomass was nearly 10 times more elastic (Figure 1a). The shear-induced fluidization of these solids was measured during the dynamic stress sweep, and it is represented by the crossover point between G’ and G”. The results show that less strain is needed to break up the 50 wt% feed into a fluid when compared with the more diluted feed (see arrows in Figure 1b), but the latter is still significantly less viscoelastic in the shear-induced liquid phase. Overall, the flow curves displayed in Figure 1c confirm this rheological behavior. For both feeds, the stress showed a plateau at lower shear rates, which is indicative of a solid with yield stress, whereas a non-Newtonian flow showed up at larger shear rates. This behavior is modeled by the Herschel-Bulkley equation, which reads as follows [37]: (1)$$\sigma =\sigma_{y}+K{\dot{\gamma }}^{n}$$
where *σ* is the shear stress, and *σ_y_* is the stress where the solid yields and then flows with the shear rate $\dot{\gamma }$ with a flow consistency coefficient *K* and a shear thinning index *n*. The fits of Equation (1) to the data plotted in Figure 1c (before the shear rate regime where instabilities showed up, see below) returned yield stresses *σ_y_* = 95 ± 33 Pa and *σ_y_* = 39 ± 25 Pa for the feeds formulated with 50 wt% and 30 wt%, respectively. More importantly, the fits returned a *K* ~ 3.5 × 10^6^ Pa·s^−n^ for the 50 wt% feed, which is four orders of magnitude larger than that for the 30 wt% feed, and a thinning index *n* = 0.9 ± 0.1 against *n* = 0.3 ± 0.1 for the 30 wt%, indicating a more viscous material but with more flow ability for the 50 wt% feed. Overall, the rheological characterization is in harmony with the expected behavior for dense suspensions with various volume fractions of solids. Figure 1 indicates that 50 mL water added to the 50 g dried biomass is sufficient to act as a binder between biomass particles for the efficient conveying of the viscoelastic feed via the screw through the squeezing chamber. Adding larger amounts of water is detrimental to *K* and *n*, which become too low to allow continuous mastication.

Figure 1 also offers all of the rheological information needed to estimate the mechanical work transmitted to the 50 wt% biomass during mastication. The mechanical work per unit volume of biomass, *W*, is given by the product of stress and strain at play in the masticating unit, that is, $W=\sigma \gamma =\sigma \dot{\gamma }t_{r}$, where σ is the shear stress and $\dot{\gamma }$ is the average shear rate in the squeezing chamber, whereas *t_r_* is the residence time in the masticating unit. With the 50 wt% biomass and a feeding rate of 10 mL per minute, the residence time of the material in the masticating unit was between 3 and 4 min, whereas the average shear rate in the squeezing unit is given by the following estimate [38]: $\dot{\gamma }=\frac{\pi \mathrm{DN}}{60h}$, where N is the screw speed (in rpm) and D the external diameter of the screw in the masticating unit. Due to the conical shape of the screw in the squeezing chamber, a maximum and a minimum shear rate can be computed as 71 s^−1^ and 46 s^−1^, respectively. Inserting these two shear rates in Equation (1), the corresponding shear stresses can be computed, and considering the residence times given above, a mechanical work *W* of the order of 1 × 10^9^ J·kg^−1^ can be estimated for 1 kg of masticated biomass. This mechanical work seems, however, overestimated when compared to those delivered by mechanical homogenizers, bead mills or ultrasonicators, which have been used to assist PHA extraction [21,22] or digestion [24]. *W* is on the order of 10^6^ J·kg^−1^ for the bead milling of NaOH digested biomasses [24], whereas mechanical works of 10^7^ and 10^9^ J·kg^−1^ were reported for mechanical homogenization and ultrasonication, respectively [39]. A possible explanation for the overestimation of *W* is given in Figure 1c. At shear rates corresponding to those taking place in the squeezing chamber of the masticating unit, the slurry with 50 wt% dried biomass actually did not flow as predicted by Equation (1). Instead, strain localization and nonhomogeneous flow, which are ubiquitous in the rheology of dense suspensions, took place as suggested by the stress plateau at larger shear rates in Figure 1c. Thus, a stress value on the order of 1 to 10 kPa should be considered to compute the mechanical work, which would then reach values between 7.5 and 126 kJ·kg^−1^ and compare to mechanical works computed for the melt mixing of, for example, polymer composites [40].

### 2.2. Effect of Biomass Mastication on PHA Recovery

Figure 2 presents a scheme summarizing the workflows used for the screening of wet biomass mastication to assist digestion with NaOH and for benchmarking with solvent extraction using CHCl_3_. Mastication was tested with the productions of samples B, C1, C5, D1 and D5, whereas samples A, E and F were produced for benchmarking.

Figure 3 compares the product recovery achieved for all processed biomasses. We can conclude that mastication had a negative impact on the recovery of products obtained by digestion because the recovery for sample B was nearly half of that for dry biomass A. However, because the recovery for sample A was larger than 1, one may suspect that the corresponding pellet was made of a significant amount of non-PHA material. Indeed, pellets containing as much as 20% impurities were recovered recently from biomasses containing 52% of PHA using a digestion protocol where much less biomass was treated in 100 mL of NaOH [28]. The authors further reported that the digestion of fresh biomass resulted in a drop in product recovery, which is in harmony with the recovery achieved using sample B, i.e., a “reconstituted fresh biomass” containing 50 wt% water. 

In order to assess whether mastication or handling of wet biomasses was responsible for the drop in product recovery, samples C1 to F were produced using the solvent extraction route. Nearly 20% recovery was achieved for sample C1, which matches the recovery obtained for the nonmasticated wet biomass E and for the masticated biomasses D1 and D5. This suggests that mastication does not impact on solvent extraction from wet biomasses. However, recovery drops below 10% for sample C5, indicating that longer wet biomass mastication in the presence of NaOH is responsible for the loss of downstreamed PHA. The mastication-assisted digestion of the wet biomass allowed downstreaming as much product as the solvent extraction from the dried biomass, as seen by comparing samples B and F in Figure 3. This suggests that for the present MMC biomass containing 50% PHA and for the large amount of biomass processed (between 10 and 20 g in 100 mL CHCl_3_ or NaOH), a better product recovery rate was achieved via digestion of wet biomasses rather than extraction of a dried biomass. This result is solid, as experiments performed with dried biomasses show that product recovery was doubled when digestion was preferred to solvent extraction with CHCl_3_ (see samples A and F). This is an important result considering the lower economic and environmental costs of NaOH digestion [36].

The levels of recovery reported in Figure 3 for extraction with CHCl_3_ are significantly lower than those reported in Table 1. This was expected given the conditions of the benchmark extraction selected here, which target higher biomass concentration in chloroform, lower temperature and shorter extraction times compared to those listed in Table 1. The inset in Figure 3 clarifies this point because it shows that adding more biomass in chloroform is detrimental to the amount of PHA recovered. This limiting factor is usually explained by the increase in the viscosity of the PHA-rich solution in CHCl_3_ [1]. Using biomass premixed with water had a negative impact on the amount of polymer recovered, which is possibly because water significantly lowers the quality of the solvent mixture (water plus chloroform) with respect to PHA. In addition, the ability of the water–chloroform mixture to make the cells’ membranes more fragile to give access to the PHA granules may have been weakened as more water was added to the biomass. However, an important result from the inset in Figure 3 is the saturation in the recovery at nearly 20 % when water was added to the biomass. Thus, when more than 5 g of dry biomass were extracted with 100 mL CHCl_3_, the water content in the biomass premix was no longer a limiting factor due to the plateauing of the recovery.

As for the mastication itself, though it did not assist the extraction in CHCl_3_, no negative impacts were shown on the recovery rates reported for biomass sonication, bead milling [21] or microwave assisted extraction [22]. This can be explained by the mechanical work associated with mastication, which is orders of magnitude smaller than the mechanical works involved during ultrasonication or bead milling (see Section 2.1). In spite of mastication not being effective in enhancing product recovery, one may question how the characteristics of recovered films and pellets compare with those of the three benchmarked products A, E and F.

### 2.3. PHA Films and Pellets Characteristics

The chemical composition of films and pellets recovered after the downstream of the biomasses was assessed with FTIR spectroscopy. The spectra are displayed in Figure 4. All samples exhibited qualitatively similar DRIFT spectra (see Figure 4a) or FTIR spectra (see Figure 4b), which show the symptomatic bands for PHA (see e.g., [41,42] and references therein), namely the functional carbonyl bonding (C=O) at 1720 cm^−1^, the ethyl (CH_2_) and the methyl (CH_3_) functional groups between 1458 cm^−1^ and 1380 cm^−1^ (which also show contributions from non-PHA residues such as collagen [20]) and the C-O-C stretching band in the region between 1279 cm^−1^ and 1182 cm^−1^ [43]. 

In addition, a band at 1540 cm^−1^ assigned to amide II and found in several MMC PHA biomasses [20,44] showed up in all spectra presented in Figure 4. This band is a clear indication of the presence of non-PHA residues in the recovered films and pellets. This band and the C=O band were indeed used to estimate the PHA content (*A_PHA_*) in biomasses, which was then validated by the PHA content measured from the TGA of the same biomasses [44]. Taking on board the approach reported by Matthew Chan et al. [44], the PHA content in the films and pellets can be computed as: (2)$$A_{PHA}=\frac{2-logT_{1720}}{4-logT_{1720}-logT_{1540}}$$
where *T*_1720_ is the intensity of the C=O band and *T*_1540_ is the intensity of the amide II band. The *A_PHA_* of all films and pellets are reported in Table 2. As expected from the recovery data (see Figure 3), the lowest *A_PHA_* values were computed from the DRIFT spectra of samples A and B. Thus, digestion with NaOH produced PHA with significant amounts of residues, which was expected because similar trends were noted for wet biomasses [35] or other biomasses with similar PHA content [26]. The FTIR analysis shows that mastication was effective in reducing the amount of non-PHA residues in the extracted films. This was not simply due to the presence of water in the biomass because sample E does not show a better *A_PHA_* than sample F. Pretreating the biomass with NaOH prior to mastication led to a significant drop in the purity of the recovered PHA films. Note here that the low *A_PHA_* values are not related to the use of DRIFT to acquire the spectra. Figure 4a shows a DRIFT spectrum of an MMC PHA produced from cheese whey using the same protocol as the one used in the present study [45]. This spectrum (labelled as CW in Figure 4a) is qualitatively similar to the other two DRIFT spectra, and the ratio *A_PHA_* computed from the bands’ intensities is 0.82. The latter is comparable to the *A_PHA_* values computed from the FTIR spectra of samples E and F and is in line with the 89 % purity estimated from the TGA of this biopolyester [45].

The purity of all films and pellets was assessed using TGA. The curves of the selected samples are shown in Figure 5. For all of the film samples produced for benchmarking, the TGA curves show a single degradation process that started at 200 °C, whereas a second process initiated above 250 °C for the pellets (see Figure 5b). Based on the DRIFT information, this second process can be assigned to the rather large amount of non-PHA residues recovered after centrifugation and the washing of the pellets. 

The purity of pellets and films is indeed inferred from the amount of weighted material (wt%) at the end of the first drop in the TGA curves, which corresponds to PHA thermal degradation. These values are reported in Table 2, whereas the inset in Figure 5b presents a graphical comparison between such purity and the ratios of *A_PHA_* computed from both FTIR and DRIFT analyses. The lower purity of the pellets correlates with lower values of *A_PHA_*, but for highly pure PHA samples (films with more than 90 wt% PHA), no correlation was found between *A_PHA_* and TGA data. This is in contrast with the linear correlations found for PHA biomasses [44], which suggests that the FTIR methodology is not valid for assessing the purity of PHA materials showing less than 10 wt% purity (films recovered in the present study). This is understandable, as the two methods rely on sensitivities towards different types of impurities: infrared spectrometry detects proteins as residues, in contrast to the residual weight made of other nonvolatile (nonorganic) impurities measured with TGA. Nonetheless, the TGA analysis confirms the information conveyed via both FTIR and DRIFT analyses. The mastication of wet MMC PHA biomasses prior to CHCl_3_ extraction enhanced the purity of the recovered polymer, whereas the use of NaOH during mastication or for biomass digestion was accompanied by a significant drop in product purity.

The TGA curves can also be used to estimate the thermal stability of the films and pellets by comparing the temperatures *T_deg_* at which the rate of weight loss is at its maximum. Temperatures *T_deg_* are listed in Table 2. Clearly, biomass digestion with NaOH gave products with bad thermal stability when compared to the benchmark, as shown by how samples A and B exhibited temperatures that were 50 to 60 °C below the *T_deg_* measured for sample F, which shows the highest *T_deg_*. Masticating with NaOH prior to extraction in CHCl_3_ did not promote the thermal stability of the recovered products because the *T_deg_* for samples C1 and C5 were in the same range as those exhibited by samples A and B. The larger amounts of non-PHA residues in these samples can explain the earlier onset for thermal degradation as reported elsewhere for MMC PHA produced from cheese whey and extracted with different amounts of impurities [45]. Smaller PHA molecular masses resulting from the chain’s hydrolysis during NaOH digestion have also been pointed out to explain weaker thermal stability [25] because lower temperatures are needed to generate volatile products from the bond-breaking of smaller PHA chains [46]. In addition, differences in the HV content of downstreamed MMC PHA could be another reason for thermal degradation at lower temperatures due to the lower melting point of HV crystals, which triggers an earlier degradation of HV-rich PHA [47]. As for the solvent extraction, the use of a wet biomass (sample E) impacts negatively on the thermal stability of the extracted PHA film. However, biomass mastication reverts this trend, as samples D1 and D5 show a *T_deg_* closer to the temperature measured for sample F.

The thermal properties measured with DSC are illustrated in Figure 6, which displays representative second heating curves. These curves present all thermal transitions reported for PHBV copolymers (see [45,48] and references therein), and the corresponding temperatures are reported in Table 2 for all samples. Starting from the lower temperatures, the glass transition ranged between *T_g_* = −14 °C and *T_g_* = −7 °C, the crystallization from the amorphous phase formed during the quenching of the melt was observed for all samples except the pellets at temperatures between *T_c_* = 43 °C and *T_c_* = 80 °C and the melting of the HV and HB crystals (2 distinct peaks) was seen at temperatures ranging from *T_m_* = 100 °C to *T_m_* = 157 °C. 

As expected from an earlier study focused on the impact of impurities on MMC PHBV’s thermal properties [45], the presence of non-PHA residues in its films and pellets shifted the glass transition and melting to lower temperatures, whereas crystallization occurred at higher temperatures. Note that the magnitudes of the shifts were consistent with the amount of impurity found in the samples to the point that the DSC curves for the pellets are featureless. In fact, pellet aging is needed for crystallization, and only the curves measured during the first heating of the pellets show the two melting processes (see the inset to Figure 6a). Overall, the temperatures listed in Table 2 suggest that the presence of impurities slows down crystallization and facilitates melting, which is probably because impurities act as defects that weaken the crystalline structure and decrease overall crystallinity [45]. However, it is known that larger HV content in the copolyester leads to a similar effect [49] and that PHBV with shorter chains equally drifts *T_m_* to lower temperatures [50]. As such, one cannot exclude the possibility that digestion and mastication with NaOH led to the selective recovery of PHBV with smaller molecular masses and containing more HV.

Based on the data reported in Figure 6, the rheological response of the molten films and pellets was measured at 170 °C. Starting with samples produced for benchmarking, Figure 7 illustrates the thermal stability of the samples extracted in chloroform with no alkali pretreatment. The time dependence of the complex viscosity |*η**| measured at 1 Hz is displayed in a semilogarithmic plot to highlight the Arrhenius type of the drop in viscosity at a later time after the initial temperature stabilization in the rheometer oven. The Arrhenius behavior of the viscosity relates to the random chain scission degradation mechanism of PHA [51], which describes a linear decrease over time of the PHA molecular mass. Because |*η**| is a power law function of the polymer macromolecular mass, an exponential decrease of |*η**| with time was recovered. 

An Arrhenius fit to the data reveals that samples D1 and D5 were slightly more stable than sample F. This was expected because TGA results showed that sample F was recovered with a few more impurities, and these are known to be detrimental to the thermal stability of PHA melts [51,52]. Sample E degraded twice as fast, which is in harmony with the TGA indicating that it started to degrade at a much lower temperature. Thus, degradation at 170 °C was sped up for sample F as compared to other films. Biomass mastication with NaOH before extraction in chloroform allowed for the recovery of a virtually thermally stable product because |*η**| was nearly constant for samples C1 and C5, whereas the viscosity of molten pellets A and B decayed at a much slower pace than sample F (see Figure 7b). This confirms that the nature of the impurities contained in the downstreamed products was more important than their amount when considering melt thermal stability [45], as all samples in Figure 7b were produced with much more impurity than sample F. In this respect, it is interesting to correlate the presence of fewer proteins in the impurities of samples D1 and D5 when compared to sample E, which otherwise shows the same level of impurity, with the improved thermal melt stability of D1 and D5. 

However, such thermal enhancement comes at a high cost as shown in Figure 8, which presents the mechanical spectra of all products at 170 °C, and in Table 2, which reports the corresponding values of |*η**| measured at 1 Hz. All of the mechanical spectra in Figure 8 were corrected for sample degradation using the Arrhenius equation and parameters computed from the fits performed in Figure 7. The mastication of the wet biomass before extraction in CHCl_3_ had a positive effect on PHA’s melt properties. The dynamic shear viscosity at 1 Hz was eight times larger after five passes in the masticating unit (compare |*η**| for samples B and C5 in Table 2), and the apparent shear thinning with frequency of the PHA extracted from the dried biomass was recovered. 

The good matches between the mechanical spectra of samples F and D5 in Figure 8a suggests that the PHA in both samples showed nearly similar molecular mass distributions. Thus, the rheological data, together with the temperatures measured with DSC and TGA, indicate that samples D1, D5 and F contained essentially the same PHA. The use of NaOH during the downstreaming of PHA had a dramatic effect on their melt rheology because a significant drop in the dynamic shear viscosity accompanied by a change from a Rouse-like to a gel-like dynamic was seen (compare the spectra of samples F and C1 in Figure 8b). A PHB produced from a biomass digested with NaOH was reported as not suitable for thermoplastic application because its molecular mass was smaller than that of a PHB extracted with chloroform from the same biomass [25]. Results for sample C1 confirm the negative effect of NaOH because the lower melt viscosity can be related to a significant drop in the PHA molecular mass. Sample C5 and the pellets extracted from the biomass digestion in NaOH exhibited a gel-like behavior. This is highlighted in the inset to Figure 8b for sample C5 with a storage modulus G’ larger than the loss modulus G” at lower frequencies. This behavior contrasts with the terminal fluid-like behavior displayed by sample F in the inset to Figure 8a. To assign the gel-like behavior to the large amount of impurities in samples C1, B and A is straightforward because they could act as reinforcing solids that fill up the melt and, at the same time, impart thermal stability to the product’s viscosity. Nonetheless, the rheology cannot discriminate between the reinforcing effect of impurities and a possible additional crosslinking between PHA chains triggered by NaOH, which further imparts an elastic behavior at lower frequencies. 

Putting the rheological characteristics presented in Figure 7 and Figure 8 into the perspective of an application such as melt extrusion, samples F, D1 and D5 showed viscoelastic behaviors which resembled those found in commercial PHB and PHBV [7,40,53] but with slightly improved thermal stability for materials produced using mastication. In contrast to this, the use of NaOH during mastication, biomass digestion, or both led to very thermally stable materials that seemed too elastic or not viscous enough to be extruded. Additional rheological testing under large deformation rates or melt extrusion trials should be performed before definitively disqualifying samples C5, B and A for such applications. However, they could be used as additives for PHA, in particular as fillers for melt reinforcement (samples A, B and C5) or as processing aids and plasticizers (samples D1 and C1) that are chemically similar to and thus compatible with other extrusion-grade PHAs.

## 3. Materials and Methods

### 3.1. MMC PHA Biomass Production

A biomass containing 50.0 ± 2.4% MMC PHA was produced from fruit waste using mixed microbial cultures in a three-stage pilot plant installed at NOVA.ID facilities. The fruit waste used for PHA production was supplied by a Portuguese juice fruit company, Sumol+Compal S.A., and it was collected from fruit pulp barrels that did not comply with the criteria to be processed by the company. This residue was characterized by having a high total COD content (174.9 ± 13 gCOD·L^−1^) and essentially being made up of readily biodegradable sugars (84.8% of soluble COD), which is an important factor to increase acidogenic potential and, thus, polymer production. The feedstock was first fermented into PHA bioprecursors in a 60 L up flow anaerobic sludge blanket (UASB) reactor as described by Matos et al. [12]. This reactor was continuously operated using an organic loading rate (OLR) of 28 ± 2 gCOD·L^−1^·d^−1^, a pH of 5.4 ± 0.3, a hydraulic retention time (HRT) of 1 d and a temperature of 30.1 ± 0.2 °C. The fermented end-stream produced was subsequently fed to two reactors, and it was mainly comprised of lactate (1.0 ± 0.3 gCOD·L^−1^), acetate (5.4 ± 0.7 gCOD·L^−1^), propionate (2.1 ± 0.2 gCOD·L^−1^), ethanol (0.8 ± 0.3 gCOD·L^−1^), butyrate (9 ± 1 gCOD·L^−1^) and valerate (2.7 ± 0.3 gCOD·L^−1^).

An aerobic 100-L sequential batch reactor (SBR) was assembled to select an MMC with a high PHA storage capacity. The culture was fed with the fermented fruit waste under 12 h feast and famine cycles, as detailed elsewhere [12]. The HRT and sludge retention time (SRT) were set at 1 and 4 days, respectively, and the OLR was maintained at 7.3 gCOD·L^−1^·d^−1^. The selective pressure favoring the enrichment of the culture in PHA-storing organisms was ensured by first applying the sequential feast–famine cycles, and second by uncoupling the nitrogen source from the carbon feeding. Semiquantitative fluorescence in situ hybridization (FISH) analyses performed throughout the SBR operation confirmed the enrichment of the community in Paracoccus, which is a well-known bacterial genus with PHA-storing capacity [12].

The PHA-enriched biomasses were finally produced in accumulation assays performed in a 60 L fed-batch reactor inoculated with the MMC selected in the second stage of the process, which was pulse-wise fed with the fermented stream produced in the acidogenic reactor. After reaching maximum PHA accumulation, sulfuric acid was added until pH 2 was reached in order to quench biological activity, the biomass was then centrifuged to remove the maximum water content, and then it was freeze-dried. The process produced 1.5 kg dried biomass with a PHA content of 50.0 ± 2.4 wt%. The polymer contained 75% HB and 25% HV, as reported elsewhere [12].

### 3.2. Biomass Mastication

A modified masticating unit (model HR1886, Philips, Eindhoven, Netherlands) was used to process the biomass. The unit consisted of a squeezing chamber made of a conical screw (minimum and maximum external diameters D of 34.5 mm and 44 mm, respectively) embedded in a conical barrel with filter (with a gap *h* between screw and barrel varying from 4.5 to 2 mm from feeding to screw tip) and terminated by a pulp outlet. The unit operated at a constant screw speed of 80 rpm. A syringe pump (model NE-300 “Just Infusion”TM, from New Era Pump Systems Inc.) operating at 10 mL per minute was used to feed the biomass in the unit. The dried MMC PHA biomass was premixed with either a fixed amount of distilled water (different premixes ranging from 30 wt% to 70 wt% biomass were tested) or with 2 M NaOH (to reach a premix of 50 wt%) before filling in a 100 mL syringe. The masticated biomass was either collected at the pulp outlet and fed again to the syringe pump for successive passes in the unit or directly used for PHA extraction and digestion. The residence time of the biomass in the masticating unit was between 2 and 3 min for the set feeding conditions and the biomass premix containing 50 wt% distilled water.

### 3.3. PHA Extraction and Digestion

For NaOH digestion, a protocol optimized elsewhere for biomasses containing different amounts (from 35 to 70%) of poly(3-hydroxybutyrate-co-3-hydroxyvalerate) (PHBV) with HV content varying from 18% to 31% [28] was employed with a modification. Instead of 2 g of biomass (dried, lyophilized or fresh [28]), 20 g of biomass (dried or masticated) was mixed under agitation with 100 mL of 0.3 M NaOH at 30 °C for 4.8 h. The mixture was then centrifuged (11,000 rpm, 10 min), and the recovered PHA-rich solid was washed several times with distilled water. The PHA-rich solid was then dried overnight at 60 °C in a ventilated oven (model FD 115, Binder GmbH, Tuttlingen, Germany), weighed to compute PHA recovery and then stored in black plastic bags until PHA characterization. Figure 2 presents a scheme summarizing the workflows used for both extraction and digestion. For PHA extraction, dried biomass (sample F), wet biomass (sample E) or the masticated biomass (with 2 M NaOH, samples C1 and C5, or with water, samples D1 and D5) were mixed with 100 mL of CHCl_3_ under agitation at 30 °C for 2 h. The PHA-rich solution was then separated from the solid biomass residues via filtration using a metal screen (1 micron). The filtrate was poured into a Petri dish and dried in a ventilated oven at 60 °C overnight. The resulting PHA films were weighed to compute PHA recovery and then stored in black plastic bags until PHA characterization. 

The product recovery, *recovery*, is defined by the following ratio:(3)$$recovery=\frac{mPRODUCT}{mBIOMASS\times 0.5}$$
where *mPRODUCT* is the weight of film or pellet recovered from the downstreaming of a dried biomass with a weight of *mBIOMASS* and initial PHA content of 50.0 ± 2.4 wt%.

### 3.4. Characterization of the Recovered Products

A TGA Q500 apparatus (from TA Instruments New Castle, DE, USA) was used to perform the thermogravimetric analysis (TGA) of extracted PHA films and digested PHA pellets. Typically, around 10 mg of film or pellets were loaded in the sample pan and were heated from 40 °C to 600 °C at a rate of 10 °C/min using nitrogen as a purge flow at a flow rate of 200 mL/min. TGA curves were acquired in triplicate. Differential scanning calorimetry (DSC) was performed with a 200 F3 Maia^®^ instrument (from Netzsch- GmbH, Kelb, Germany). The samples were placed in a closed aluminum pan and subjected to three scans. A first heating (25–180 °C at 10 °C/min) was performed to erase the thermal history of the sample. Then, a cooling ramp was conducted at a rate of −10 °C/min down to −25 °C, where the sample was maintained for 5 min. Finally, a second heating was applied under the same conditions as the first heating. The second heating was used to determine all DSC output gathered in Table 2. All DSC experiments were done under nitrogen purging. The FTIR spectra of film samples and powdered pellets were acquired with an FTIR Spectrometer (Spectrum100, PerkinElmer Inc., Waltham, MA, USA) equipped with an attenuated total reflectance (ATR) accessory for measuring films and a diffuse reflectance (DRIFT) accessory for measuring the pellets. For each sample, 32 scans were recorded between 4000 and 400 cm^−1^ and averaged to give spectra with a resolution of 1 cm^−1^.

### 3.5. Rheology

A stress-controlled rotational rheometer (ARG2, TA Instruments, New Castle, DE, USA) was used with 25 mm parallel plates and a gap varying between 0.55 and 1.02 mm for the characterization of biomass premixes in water (slurries). All tests were performed at room temperature. Right after loading a slurry between the plates of the rheometer, a dynamic time sweep using a small amplitude oscillatory shear (SAOS) with a frequency of 1 Hz and an amplitude of 0.05% was performed for 5 min to monitor the structural recovery of the slurry after loading. A stress sweep in oscillatory mode at 1 Hz was then performed from 100 to 5000 Pa to determine the regime of linear viscoelasticity (LVE). In a second set of experiments, another sample was loaded, and the same dynamic time sweep was performed as above. This was followed by a frequency sweep from 100 Hz down to 0.01 Hz performed with a SAOS of 0.05%, which belongs to the LVE regime determined in the previous experiment. Finally, a third experiment was carried out with a third sample. After performing the same dynamic time sweep, a flow curve was measured by ramping the steady shear stress from 10 Pa to 5000 Pa, applying each stress during 10 s and reading the shear rate during the last two seconds of the applied stress to compute an average shear rate. The experimental protocol used for characterizing the PHA films and pellets at 170 °C with 40 mm parallel plates encompassed a dynamic time sweep (at 1 Hz with a SAOS varying from 0.1% to 10% depending on the LVE regime of the tested sample) applied for 8 min to monitor the thermal degradation of the film or pellets after loading between the preheated plates. Then, a frequency sweep was performed from 100 to 0.03 Hz with a SAOS varying between 0.1% and 10% depending on the PHA sample. The LVE regime was ensured by monitoring the sinusoidal traces of both dynamic stress and strain recorded online by the rheometer. The frequency sweep was repeated several times to further probe the thermal degradation of the samples.

## 4. Conclusions

The masticating device used in the present study was not effective in assisting the digestion of a wet mixed microbial biomass in NaOH for the recovery of more PHA. The recovered products contain significant amounts of impurities and thus may only find application as rheological modifiers or thermal stabilizers for PHA. In spite of the too-small improvement in the recovery of pure PHA with CHCl_3_, the new tested continuous mechanical pretreatment is attractive when large amounts of wet biomass are processed. Indeed, the product recovered with mastication shows better melt stability and melt viscoelasticity. There is still ample room for exploring routes to improve the mastication of wet PHA biomasses in future studies. Mastication parameters and unit designs can be changed to increase mechanical energy and allow the processing of biomasses containing more water. It will be of interest to check whether mastication can boost both PHA recovery and thermal properties for biomasses with higher PHA content. Alternatively, testing biomasses with lower PHA content and higher water content with a modified prototype enabling biomass dewatering before mastication will be challenging, as such feeds are known to be very difficult to downstream. Finally, future studies should also screen biomasses containing other types of PHA, e.g., PHBV with different HV contents, and extend to economically and ecologically attracting solvents to confirm the results achieved here with CHCl_3_. 

## Figures and Tables

**Figure 1 molecules-28-00767-f001:**
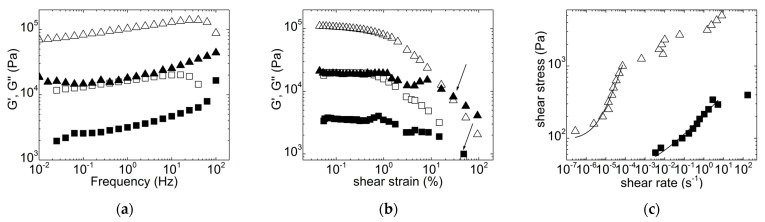
Mechanical spectra (**a**), dynamic stress sweeps (**b**) and flow curves (**c**) performed on biomass premixes with distilled water: 50 g with 50 mL (triangles) and 30 g with 70 mL (squares). In (**a**,**b**): G’ is displayed as open symbols; G” is displayed as filled symbols. Arrows in (**b**) indicate the strains where slurries are fluidized, whereas lines in (**c**) are fits of Equation (1) to the data.

**Figure 2 molecules-28-00767-f002:**
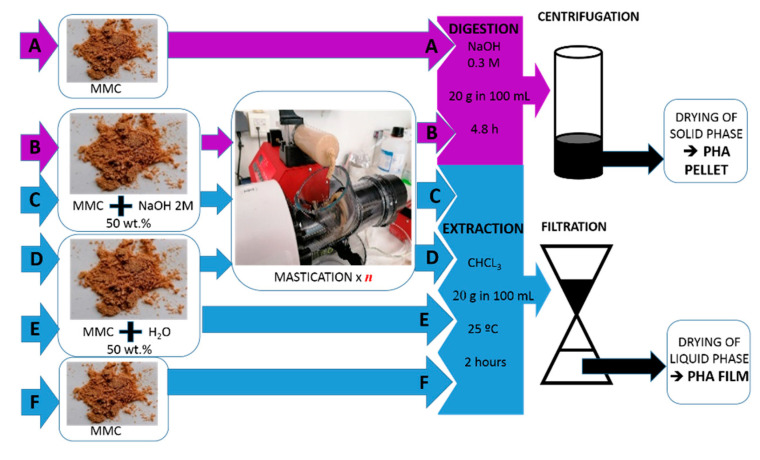
Flow chart of the downstreaming protocols used for digestion with 0.3 M NaOH or extraction with CHCl_3_. The dried biomass (MMC) containing 50 wt% PHA was processed using routes A to F, yielding eight samples in the form of pellets (digestion, samples A and B) or films (extraction, samples C to F). Samples C1 and D1 were masticated with *n* = 1, whereas samples B, C5 and D5 were masticated with *n* = 5.

**Figure 3 molecules-28-00767-f003:**
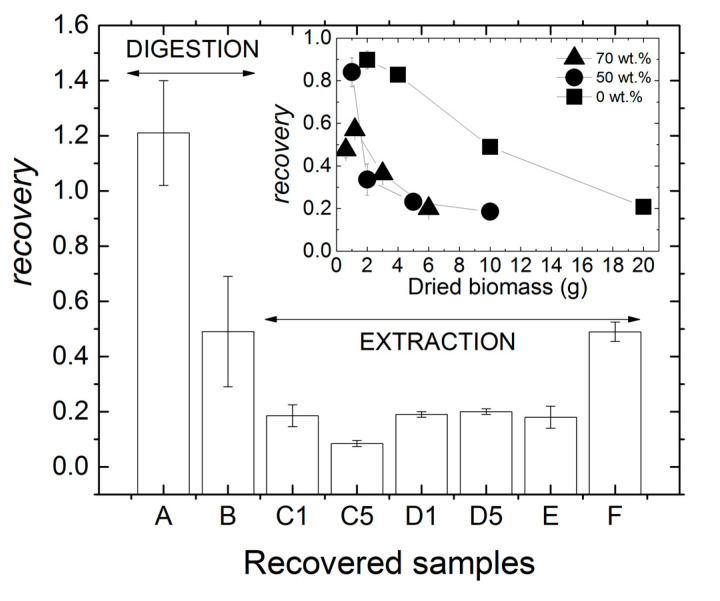
Product recovery achieved for all processed biomasses. Samples produced for benchmarking are: A for digestion with NaOH, and E and F for extraction in CHCl_3_. Inset: product recovery as a function of the dried biomass weight used for the extraction with CHCl_3_ of MMC PHA.

**Figure 4 molecules-28-00767-f004:**
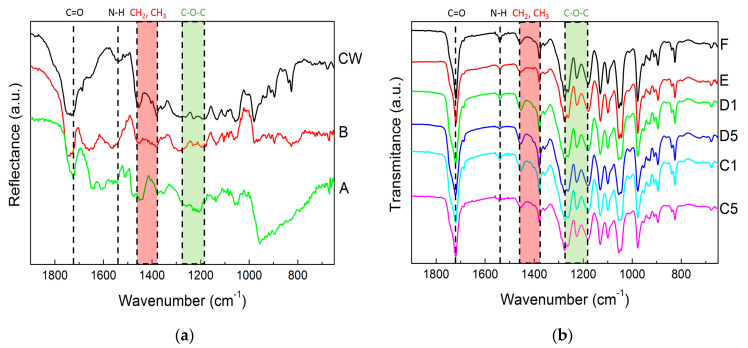
DRIFT spectra of pellets recovered from the digestion with NaOH compared to an MMW PHA produced from cheese whey and digested with NaClO (CW) (**a**) and FTIR spectra of films recovered from extraction in CHCl_3_ (**b**).

**Figure 5 molecules-28-00767-f005:**
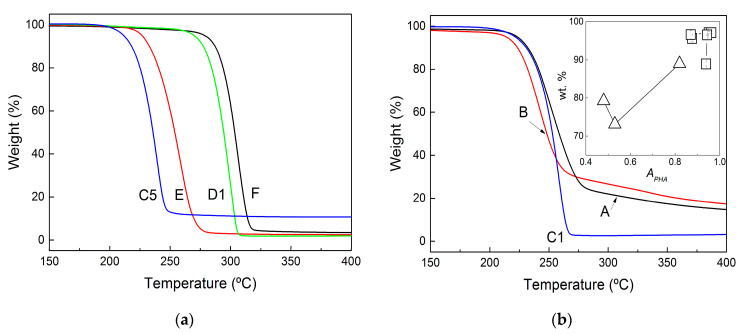
TGA curves of recovered films extracted with chloroform (**a**) and of pellets recovered after biomass digestion with NaOH (**b**). The inset in (**b**) relates the PHA purity inferred from both FTIR (squares) and DRIFT (triangles) analyses to the purity measured in weight percent (wt%) by using TGA.

**Figure 6 molecules-28-00767-f006:**
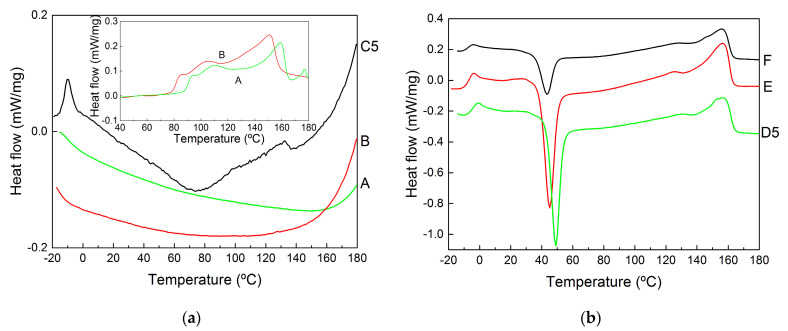
DSC curves (measured during the second heating and vertically shifted for better visualization of the results) of samples recovered from the biomass digestion in NaOH (**a**) or extracted with CHCl_3_ (**b**). The inset in (**a**) shows the DSC curves acquired during the first heating of the pellets.

**Figure 7 molecules-28-00767-f007:**
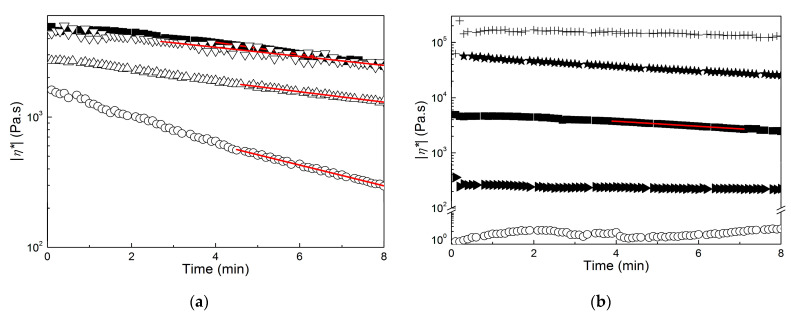
Time dependence of the dynamic shear viscosity |*η**| measured at 1 Hz and 170 °C. (**a**): Sample F (squares), sample E (circles), sample D1 (up triangles) and sample D5 (down triangles); (**b**): sample F (squares), sample C1 (circles), sample C5 (right triangles), sample B (stars) and sample A (crosses). Lines are Arrhenius fits to the data.

**Figure 8 molecules-28-00767-f008:**
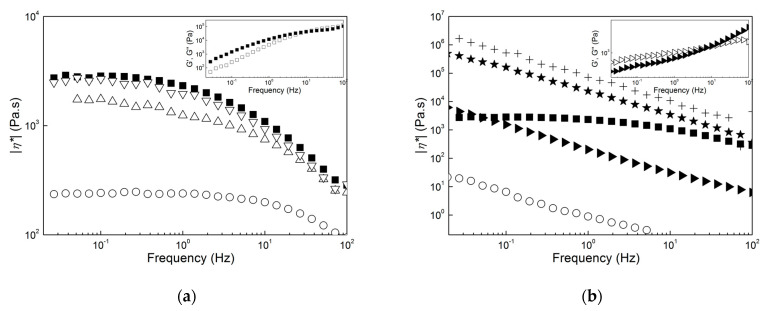
Mechanical spectra (dynamic shear viscosity |*η**| as a function of frequency) measured at 170 °C. (**a**): Sample F (squares), sample E (circles), sample D1 (up triangles) and sample D5 (down triangles); (**b**): sample F (squares), sample C1 (circles), sample C5 (left triangles), sample B (stars) and sample A (crosses). Inset in (**a**): mechanical spectrum (storage modulus G’ (solid symbols) and loss modulus G”(empty symbols) as a function of the frequency) of sample F. Inset in (**b**): mechanical spectrum of sample C5.

**Table 1 molecules-28-00767-t001:** Product recovery and PHA purity reported for the MMC PHA biomass downstreaming with chlorinated solvents or NaOH digestion. The PHA content (PHA) in the initial biomass is given in weight percent (wt%), and the proportion of 3-hydroxyvalerate (HV) in the recovered poly(3-hydroxybutyrate-co-3-hydroxyvalerates) (PHBV) is given in %. The concentration c of dry biomass in chloroform is given in weight per volume percent (% *w*/*v*) in the corresponding solution (with CHCl_3_, CH_2_Cl_2_ or NaOH with the corresponding molar concentration) and was computed from the reported type of biomass: dried (*d*), lyophylized (*l*) or wet (*w*), as computed from the referenced water content. The parameters of the downstreaming process are temperature (*T* in °C) and time (*t* in hours). PHA recovery and purity are given in %. “n.a.” indicates data are not available.

PHA (wt%)	HV	Extraction			Recovery	Purity	Reference
		*c* (% *w*/*v*)	*T*	*t*			
Chlorinated solvent extraction							
70	31	4 (*l*) CHCl_3_	80	48	82 ^a^	95 ^d^	[28]
66	18	2 (*l*) CHCl_3_	80 (Soxhlet)	24	66 ^b^	91 ^d^	[29]
40	24	2.5 (*l*) CH_2_Cl_2_	50	4	52 ^a^	94	[11]
25	5.6	n.a. (CH_2_CL_2_ + water)	61	0.5	18–30 ^a^	n.a.	[20]
72	0	1 (*l*) CH_2_Cl_2_	25	12	56 ^a^	98 ^d^	[25]
72 ^e^	66	2 ^e^ (*w*) CHCl_3_ Sonication Beads mill Mechanical homogeneisation	61	2	31 ^e^ 7 ^e^ 24 ^e^ 44 ^e^	94.5 ^d^ 90.6 ^d^ 93.3 ^d^ 88.3 ^d^	[21]
n.a.	17.5	5 (*l*) CHCl_3_ Microwave	100	2	36 ^c^ 1.5–38 ^c^	<96 ^d^ >96 ^d^	[22]
50	12	2.5 (*l*) CHCl_3_	38	72	63.5 ^a^	n.a.	[30]
38.7–56.1	7.7–20.2	2.9–7.1 (*d*) CHCl_3_	61 (Soxhlet)	24	75.4–97.9 ^f^	83–100	[31]
n.a.	4–5	5 (*d*) CHCl_3_	60	2	32 ^a^	>99 ^d^	[32]
54	0	6.7 (*d*) CHCl_3_	60	2	69 ^a^	100 ^d^	[33]
32	50	1 (*l*) CHCl_3_	60 (reflux)	1	37.5 ^b^	82.5	[34]
56 62	17 11	n.a. (*d*) CHCl_3_ n.a. (*w*) CHCl_3_	61 (Soxhlet)	8	96 ^a^	98	[35]
50	25	20 (*d*) CHCl_3_ 10 (*w*) CHCl_3_ 10 (*w*) CHCl_3_ and mastication	30	2	46.7 ^a^ 17.4 ^a^ 7.6–19.4 ^a^	95.6 96.6 88.9–97.1	This study
NaOH digestion							
70	31	2 (*d*) in 0.3 M	30	4.8	100 ^a^	100	[28]
65–73	0	2 (*w*) in 0.02 M to 1 M 2 (*l*) in 0.2 M	30	0.3–3 1	86–98.5 ^a^ 95.5 ^a^	77–92 96	[25]
46	13	n.a. in 1 M	n.a.	3–24	80–87 ^a^	75	[26]
64	13	2.5 (*l*) in 0.3 M	30	1	90 ^a^	76	[27]
50	25	20 (*d*) in 0.3 M 10 (*w*) in 0.3 M and mastication	30	4.8	95 ^a^ 35.8 ^a^	79 73.1	This study

^a^: pure PHA recovered over the PHA content in the biomass. ^b^: pure PHA recovered over the biomass treated. ^c^: extracted product over the biomass treated. ^d^: after precipitation in nonsolvent. ^e^: fresh (wet) biomass weight; PHA content in biomass and PHA recovery were computed from the ATR-FTIR analysis of the biomass after extraction. ^f^: computation not detailed.

**Table 2 molecules-28-00767-t002:** Samples produced from the downstreaming of corresponding biomasses with parameter *n*, with their purities quantified by FTIR (*A_PHA_*) and TGA (wt%) as well as their melt properties (temperatures *T_deg_*, *T_g_*, *T_c_* and *T_m_* in °C, and dynamic shear viscosity |*η**| in Pa·s).

Biomass	*n*	Sample	Purity		Melt Properties				
			*A_PHA_*	wt%	*T_deg_*	*T_g_*	*T_c_*	*T_m_*	|*η**|^d^
Digestion with NaOH									
A	0	A	0.48 ^a^	79.2 ± 1.7	253.8 ± 1.4	-	-	110 ^c^/159 ^c^	71,560
B	5	B	0.53 ^a^	73.1 ± 1.9	241 ± 1	-	-	106 ^c^/151 ^c^	23,350
Extraction in CHCl_3_									
C	1	C1	0.94	96.5 ± 0.1	258.0 ± 1.3	−8.5	80.4	110/133	0.9
C	5	C5	0.94	88.9 ± 0.5	238.5 ± 1.6	−14 ^b^	74 ^b^	100 ^b^/132	204
D	1	D1	0.95	97.1 ± 0.3	299.1 ± 0.6	−7.6	50.2	129/156	1241
D	5	D5	0.96	97.1 ± 0.3	301.7 ± 1.1	−7.2	49.2	130/157	1954
D	1	D1	0.95	97.1 ± 0.3	299.1 ± 0.6	−7.6	50.2	129/156	1241
E	0	E	0.87	96.6 ± 1.0	258.4 ± 0.9	−9.6	45.2	124/156	239
F	0	F	0.88	95.6 ± 0.7	306.0 ± 2.5	−9.9	43.5	128/156	2320

^a^: Computed from DRIFT spectra. ^b^: Errors are 1, 2 and 5 °C for Tg, Tc and first Tm, respectively, whereas errors are 0.5 °C for all other samples. ^c^: Measured from the first heating of the pellets, with an error of 1 °C. ^d^: Measured at 170 °C and a frequency of 1 Hz.

## Data Availability

Data is contained within the article.

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
