# Peer review of "Can Biomass Mastication Assist the Downstreaming of Polyhydroxyalkanoates Produced from Mixed Microbial Cultures?"

_molecules, 2023, doi:10.3390/molecules28020767_

Round 1

Reviewer 1 Report

This work explored mastication as a new route to enhance the fragility of the cells, thereby facilitating the digestion of the non-PHA material by NaOH. The structure of the paper was reasonable, and the prospect was put forward based on the research status. Some issues need to be solved and minor revision is necessary for publication on Molecules. The detailed comments are as follows:

1. Materials and methods should be put before results and discussion.

2. Reference should be added to equation 1.

3. The abscissa of Figure 4a is incomplete.

4. The ordinates of Figure 4a and Figure 4b shall be uniform.

5. The format should be uniform, such as a space between numbers and units.

6. The conclusion is too long, and the author should be simplified and condensed into one paragraph.

7. A total production cost of the method in this work should be added, especially the cost comparison with other methods reported in the references.

Author Response

  1. Materials and methods should be put before results and discussion.

ANSWER: we understand that the reading of the paper would be easier following this reviewer’s comment. However, we did stick to the guidelines and template set by the journal. We thus propose to keep the outline of the journal and which is followed by the majority of the research articles published in Molecules. If the Editorial office decide otherwise, we will then reorganize the paper’s structure as per the reviewer’s indication.

  1. Reference should be added to equation 1.

ANSWER: the following reference to the original paper introducing this equation has been added to address this comment:

Herschel, W.H.; Bulkley, R. (1926), "Konsistenzmessungen von Gummi-Benzollösungen", Kolloid Zeitschrift, 39 (4): 291–300, doi:10.1007/BF01432034, S2CID 97549389.

  1. The abscissa of Figure 4a is incomplete.

ANSWER: we now provide a new figure with abscissa similar to the one used in Figure 4b.

  1. The ordinates of Figure 4a and Figure 4b shall be uniform.

ANSWER: we understand that since both figures show infrared spectra, it will be more convenient to display transmittance or absorption for both. However, we kindly draw the reviewer’s attention to the fact that diffuse reflectance is reported in Figure 4a, as pellets were measured, in contrast to the transmittance measured with films used in transmission for Figure 4b. As diffuse reflectance and transmission deal with different material-light interactions, a simple conversion from reflection to absorbance and eventually transmittance does not exist. Though both DRIFT and FTIR bring semi-quantitative chemical information, these are not quantitatively the same, and similar ordinates cannot be plotted in Figure 4. However, non-dimensional ratios of band intensity can be comnpared.

  1. The format should be uniform, such as a space between numbers and units.

ANSWER: the format of the paper has been carefully corrected, see track changes in the revised manuscript.

  1. The conclusion is too long, and the author should be simplified and condensed into one paragraph.

ANSWER: we significantly reduced the conclusion which now reads as: “The masticating device used in the present study was not effective in assisting the digestion of a wet mixed microbial biomass in NaOH for the recovery of more PHA. The recovered products contain significant amounts of impurities and thus may only find application as rheological modifiers or thermal stabilizers for PHA. In spite of the too small improvement in the recovery of pure PHA with CHCl3, the new tested continuous mechanical pre-treatment is attractive when large amounts of wet biomass are processed. Indeed, the product recovered with mastication shows better melt stability and melt viscoelasticity. There is still ample room for exploring routes to improve the mastication of wet PHA biomasses in future studies. Mastication parameters and unit designs can be changed to increase the mechanical energy and allow the processing of biomasses containing more water. It will be of interest to check whether mastication can boost both PHA recovery and thermal properties for biomasses with larger PHA contents. Alternatively, testing lower PHA contents and higher water contents with a modified prototype enabling biomass dewatering before mastication will be challenging as such feeds are known to be very difficult to downstream. Finally, future studies should also screen biomasses containing other types of PHA, e.g. PHBV with different HV contents, and extend to economically and ecologically attracting solvents to confirm the results achieved here with CHCl3.”.

  1. A total production cost of the method in this work should be added, especially the cost comparison with other methods reported in the references.

ANSWER: such cost analysis is indeed imperative whenever the concept of the method is established. This is not the case here, as shown in the revised title of the paper which brings a question mark (it now reads as “Can biomass mastication assist the down streaming of polyhydroxyalkanoates produced from mixed microbial cultures?”), as well as in the first line of the revised conclusion (see answer to above comment). As such, a careful cost assessment including a benchmarking with similar studies will be premature at this stage, since many mastication parameters and other biomasses need to be tested first, before a TRL 3 or 4 is reached. In addition, a cost comparison with the referenced methods will be impossible since none of the studies referenced table 1 performed any production cost analysis. It seems thus that production cost and/or LCA is not yet state-of-the-art-practice for these research topics. We do share the idea of this reviewer that such analysis should be compulsory, but of course only when all parameters for the production cost and LCA are available, not when a concept is simply tested.

Reviewer 2 Report

1. Page 1, Line 38: Please add reference for this statement ‘…PHA is a more attractive bio-sourced polyester as it naturally biodegrades in land field or the ocean [Add Ref]’. Add reference of ‘Polyhydroxyalkanoates: Production and Biodegradation - A Review. Encyclopedia of Materials: Plastics and Polymers. doi:10.1016/B978-0-12-820352-1.00265-0’.

2. Page 1, Line 40: Please add reference for this statement ‘…applications such as plastic packaging [Add Ref]. Add reference of ‘A Review of the Applications and Biodegradation of Polyhydroxyalkanoates and Poly(lactic acid) and Its Composites. Polymers 2021, 13, 1544. https://doi.org/10.3390/polym13101544’.

3. Page 14, Line 476: Please provide detailed elaboration regarding the raw material of the fruit waste utilized in this study. Also, please provide the type of mixed microbial culture (majority/dominantly consists of what strain?) utilized for PHA production. The authors should select and use the microbes that can accumulate high PHA content.

4. Section materials and methods: The authors are highly suggested to provide the instrument model, city, country (maker) for all equipment’s used in this study.

5. Page 7, Figure 3: Unit for recovery?

6. Page 13: Please separate the text in this page by adding paragraph.

7. Results and discussion: Mastication process or technique utilized in this study contain drawbacks as indicated by low PHA recovery. The authors utilized biomass contained 50 % of PHA on a dry basis. It is highly suggested to utilized biomass with high PHA% for testing the mastication process or technique, for example, 80-90% accumulated PHA.

<<END>>

Author Response

  1. Page 1, Line 38: Please add reference for this statement ‘…PHA is a more attractive bio-sourced polyester as it naturally biodegrades in land field or the ocean [Add Ref]’. Add reference of ‘Polyhydroxyalkanoates: Production and Biodegradation - A Review. Encyclopedia of Materials: Plastics and Polymers. doi:10.1016/B978-0-12-820352-1.00265-0’.

ANSWER: the proposed reference is now added.

  1. Page 1, Line 40: Please add reference for this statement ‘…applications such as plastic packaging [Add Ref]. Add reference of ‘A Review of the Applications and Biodegradation of Polyhydroxyalkanoates and Poly(lactic acid) and Its Composites. Polymers 2021, 13, 1544. https://doi.org/10.3390/polym13101544’.

ANSWER: indeed, reference [3] in the original paper does this job. However, the reference was not inserted at the end of the statement. We apologize for this flaw. The reference is now shifted accordingly and the revised line now reads: “However, by 2021, PHA production is a tenth of PLA production, and remains a very marginal player in applications such as plastic packaging [4].”.

  1. Page 14, Line 476: Please provide detailed elaboration regarding the raw material of the fruit waste utilized in this study. Also, please provide the type of mixed microbial culture (majority/dominantly consists of what strain?) utilized for PHA production. The authors should select and use the microbes that can accumulate high PHA content.

ANSWER: the corresponding experimental section has been corrected accordingly to give detail and now reads as

“A biomass containing 50.0 ± 2.4 % MMC PHA was produced from fruit waste using mixed microbial cultures in a three-stage pilot plant installed at NOVA.ID facilities. The fruit waste used for PHA production was supplied by a Portuguese juice fruit company, Sumol+Compal S.A., and it is collected from fruit pulp barrels that do not comply with the criteria to be processed by the company. This residue is characterized by having a high total COD content (174.9 ± 13 gCOD.L-1), es-sentially made up of readily biodegradable sugars (84.8 % of soluble COD), which is an important factor to increase the acidogenic potential and thus the polymer production. The feedstock was first fermented into PHA bioprecursors in a 60- L upflowup flow an-aerobic sludge blanket (UASB) reactor as described by Matos et al. [12]. This reactor was continuously operated using an organic loading rate (OLR) of 28 ± 2 gCOD.L-1.d-1, pH at 5.4 ± 0.3, an hydraulic retention time (HRT) of 1 d and temperature of 30.1 ± 0.2 ºC. The fermented end-stream produced was subsequently fed to two reactors and it was mainly comprised of lactate (1.0 ± 0.3 gCOD.L-1), acetate (5.4 ± 0.7 gCOD.L-1), propionate (2.1 ± 0.2 gCOD.L-1), ethanol (0.8 ± 0.3 gCOD.L-1), butyrate (9 ± 1 gCOD.L-1) and valerate (2.7 ± 0.3 gCOD.L-1).

An aerobic 100-L sequential batch reactor (SBR) was assembled to select a MMC with a high PHA-storage capacity. The culture was fed with the fermented fruit waste under 12- h feast and famine cycles, as detailed elsewhere [12]. The HRT and a sludge retention time (SRT) were set at 1 and 4 days, respectively, and the OLR was maintained at 7.3 gCOD.L-1.d-1. The selective pressure favoring the enrichment of the culture in PHA-storing organisms was ensured by first, applying the sequential feast-famine cycles, and second by uncou-pling the nitrogen source from the carbon feeding. Semi-quantitative fluores-cence in situ hybridization (FISH) analyses performed throughout the SBR operation con-firmed the en-richment of the community in Paracoccus, which is a well-known bacterial genus with PHA storing capacity [12].”.

  1. Section materials and methods: The authors are highly suggested to provide the instrument model, city, country (maker) for all equipment’s used in this study.

ANSWER: such detail is now provided when available, see pages 14-16

  1. Page 7, Figure 3: Unit for recovery?

ANSWER: as shown in equation (3) page 15, recovery is a ratio of 2 weights. This non-dimensional ratio being directly plotted in Figure 3 (not in %), it has no unit.

  1. Page 13: Please separate the text in this page by adding paragraph.

ANSWER: done.

  1. Results and discussion: Mastication process or technique utilized in this study contain drawbacks as indicated by low PHA recovery. The authors utilized biomass contained 50 % of PHA on a dry basis. It is highly suggested to utilized biomass with high PHA% for testing the mastication process or technique, for example, 80-90% accumulated PHA.

ANSWER: We appreciate the suggestion. Indeed we pointed this idea in the conclusion with the following specific lines for future studies: “It will be of interest to check whether mastication can boost both PHA recovery and thermal properties for biomasses with larger PHA contents”.

Reviewer 3 Report

1.The manuscript entitle "Synthesis, Characterization and Application of Polymer-Based Materials" is well written organized and have interesting results the article seems useful for the readers of the journal in broad and to the readers of the field in specific. However the following points should be addressed before publication:

1Title: the title of the manuscript seems reviews article in first glance, it should be more specific and self explanatory

   Introduction: introduction is well written, plenty of research in the related field is available in recent days so author should cite the more recent work.

   Experimental: the author has adopt state of the art procedures, however there are some grammatical errors which should check very carefully.

  Results and discussion: As commented above the author should support their work with the recent work

  Conclusion: conclusion should be more self explanatory and to the point, so author should revisit it more carefully

  References: The author should cite more recent references \      

   With these minor comments, i feel pleasure to recommend this manuscript for publication

5.       

Author Response

The manuscript entitle "Synthesis, Characterization and Application of Polymer-Based Materials" is well written organized and have interesting results the article seems useful for the readers of the journal in broad and to the readers of the field in specific. However the following points should be addressed before publication:

1Title: the title of the manuscript seems reviews article in first glance, it should be more specific and self explanatory

ANSWER: a new title is proposed to address this specific comment: “Can biomass mastication assist the down streaming of polyhydroxyalkanoates produced from mixed microbial cultures?”.

   Introduction: introduction is well written, plenty of research in the related field is available in recent days so author should cite the more recent work.

ANSWER: as mentioned in the introduction, the topic of the paper is focused on benchmarking with extraction in chloroform and digestion with NaOH. Also, we do not extend to other PHA down streaming, and there is no much recent literature on NaOH and chloroform as far as we are aware. This is stated in the introduction, but we would highly appreciate if this reviewer indicates specific references that we would have missed.

   Experimental: the author has adopt state of the art procedures, however there are some grammatical errors which should check very carefully.

ANSWER: done, see the modified section.

  Results and discussion: As commented above the author should support their work with the recent work

ANSWER: results on the effect of mastication are compared with works reported in table 1 and citing 14 references. As mentioned in the answer to the second comment for the introduction, we are not aware of other published works to be used for benchmarking our results. However, among the 14 references cited, 4 were published between 2009 and 2015, whereas the remaining 10 cover the period 2020-2022. We believe this can be considered as recent. The discussion of PHA characteristics mainly relies on state-of-the-art concepts on polymer science, and on seminal original papers on PHA fundamental characteristics. More recent papers on PHA characterization actually refer to this literature which we think fair to cite in the first place.

  Conclusion: conclusion should be more self explanatory and to the point, so author should revisit it more carefully

ANSWER: following also a similar comment from reviewer 1, the conclusion has been modified accordingly and now reads as: “The masticating device used in the present study was not effective in assisting the digestion of a wet mixed microbial biomass in NaOH for the recovery of more PHA. The recovered products contain significant amounts of impurities and thus may only find application as rheological modifiers or thermal stabilizers for PHA. In spite of the too small improvement in the recovery of pure PHA with CHCl3, the new tested continuous mechanical pre-treatment is attractive when large amounts of wet biomass are processed. Indeed, the product recovered with mastication shows better melt stability and melt viscoelasticity. There is still ample room for exploring routes to improve the mastication of wet PHA biomasses in future studies. Mastication parameters and unit designs can be changed to increase the mechanical energy and allow the processing of biomasses containing more water. It will be of interest to check whether mastication can boost both PHA recovery and thermal properties for biomasses with larger PHA contents. Alternatively, testing lower PHA contents and higher water contents with a modified prototype enabling biomass dewatering before mastication will be challenging as such feeds are known to be very difficult to downstream. Finally, future studies should also screen biomasses containing other types of PHA, e.g. PHBV with different HV contents, and extend to economically and ecologically attracting solvents to confirm the results achieved here with CHCl3.”.

  References: The author should cite more recent references \   

ANSWER: see above answers which addressed this specific point. As a last note, we emphasize here that 26 of the 53 references cover the period 2019-2022.